# Using the Mechanisms of Action Involved in the Pathogenesis of Androgenetic Alopecia to Treat Hair Loss

**DOI:** 10.3390/ijms262110712

**Published:** 2025-11-03

**Authors:** Houfar Sekhavat, Sara Bar Yehuda, Satish Asotra

**Affiliations:** Triple Hair Inc., Dieppe, NB E1A 7R8, Canada; houfar@hotmail.com (H.S.); drasotra@icloud.com (S.A.)

**Keywords:** androgenetic alopecia, minoxidil, finasteride, latanoprost, DHT, Wnt/β-catenin pathway, VEGF, cell cycle regulators, IGF-1, Ca^2+^/K channels

## Abstract

Androgenetic alopecia (AGA) is the most common type of baldness, characterized by progressive miniaturization of the hair follicle and eventually atrophy. Both genetic and androgenic factors play definite roles in the pathophysiology of the disease, including androgens and growth factors, which induce a crosstalk between the dermal papilla and the hair follicle cells. The goal of AGA treatments is to prevent the hair miniaturization process; however, currently there are only two FDA-approved medications to treat AGA: topical Minoxidil (5% and 2%) for men and women, and oral Finasteride (1 mg tablets—Proscar and Propecia) for men. Nevertheless, these are costly, require lifelong treatment, and may have side effects. Thus, there have been many attempts to develop drugs that can harness the mechanisms controlling the pathogenesis of AGA. These pharmacological therapies might achieve more targeted and effective treatment for the disease. In this review, we present various treatments that have demonstrated their ability to induce hair growth by controlling the pathophysiological mechanisms involved in the development of AGA. Interestingly, treatment by a combination of some drugs has resulted in better outcomes than each of the drugs alone, hence demonstrating the advantage of activating different molecular mechanisms simultaneously.

## 1. Introduction

Hair growth on the head has an important role in heat regulation by the human body, either by providing insulation or by cooling through evaporating sweat from soaked hair. Head hair also protects the scalp against ultraviolet radiation by acting as a physical barrier that serves as a sunscreen [1,2]. In addition, hair makes an important contribution to the appearance of the human body, playing a vital role in social and sexual communications [3].

Hair comprises two distinct structures: the hair follicle (beneath the skin, including the capillary, the hair papilla, the hair bulb, the hair follicle, the hair bulge, the arrector pili muscle, and the sebaceous gland) and the shaft of the hair (hard filamentous part that extends above the skin’s surface). The dermal papilla, which is part of the hair follicle and serves as the main mesenchymal component, has an important role in the induction of new hair follicle formation and maintenance of hair growth [4]. The blood vessels from the dermal papillae nourish all hair follicles and offer nutrients and oxygen to epidermal cells in the lower layers. The bulb, which is located in the dermis, maintains stem cells that re-grow the hair once strands fall out and is also recruited to repair the skin after a wound [5]. Hair growth initiates at the base of the hair follicle in the hair bulb. Cell proliferation occurs in the matrix of the hair bulb, after which the keratinocytes move up into the follicle, differentiating into the layers of the hair and its surrounding sheaths [6] (Figure 1).

The human hair cycle consists of three different stages: Anagen—the growth phase, accompanied by extensive mitotic activity, which lasts between 2 and 7 years. About 90% of hair is at this stage; catagen—the regression phase, in which the follicular keratinocytes undergo apoptosis, which lasts for about 2 weeks. Approximately, one to two percent of hair is at this stage; telogen—the quiescent inactive phase, lasting for about 3 months. Around 9% of hair is at this stage. The cyclic changes from anagen to telogen occur via remodeling of both the epithelial and dermal components of hair follicles during the catagen phase. Throughout this process, hairs are shed until re-entry into anagen to generate a new hair shaft in the existing follicle. In the telogen phase, the old hair is lost, but new hair begins to grow in the same hair follicle, and the anagen growing cycle begins again. For this to happen, the dermal papilla cells support increased cell division and growth rate, which requires a good supply of nutrients and a toxin-free environment for the growing cells. If these requirements are not fulfilled, the follicles will remain in the telogen phase [8,9,10].

Hair loss, known as alopecia areata, is a dermatological disorder that has been recognized for more than a thousand years as a common problem in cosmetics as well as in primary health care practices. It is seen all over the world and affects approximately 2% of the world’s population [11].

There are two factors that usually lead to hair loss. The first factor is the shortening of the anagen phase, which is manifested by shorter hairs, hair shaft loss, and an increased number of hairs in the telogen phase. The second factor is the small size of the dermal papilla or the hair matrix, leading to hair modification both in diameter and appearance. Consequently, thick, pigmented terminal hair turns into thin and depigmented vellus hair [9,11,12].

## 2. Androgenetic Alopecia

Androgenetic alopecia (AGA) is the most common type of baldness, and its prevalence is rising with increasing life expectancy. It affects at least 80% of men and 50% of women by age 70, with incidence increasing with age. The disease is characterized by shortening of the anagen phase of the hair cycle and prolongation of the telogen phase, leading to progressive miniaturization of the hair follicle. This chain of events results in increased shedding of short-lived hairs and the formation of shorter, finer vellus hairs that eventually atrophy [13,14].

It is well established that hair loss and baldness may induce deep emotional stress [5] and reduced self-esteem. It has been found that alopecia is associated with feelings of anxiety, depression [6], anger, reduced satisfaction with body image, low self-esteem, and lack of self-confidence [15,16,17]. Social dysfunction—which may be experienced by impaired social interactions with the opposite gender, fear of appearing in public, and discomfort during interactions with strangers—was reported by individuals suffering from various degrees of hair loss [17,18]. It has been reported that patients with AGA believe their physical appearance due to the disease (the hair loss and baldness) causes distress and feelings of depression, anxiety, and social phobia [19]. In a cross-sectional, observational study, 133 AGA patients were interviewed in order to evaluate the prevalence of depression, levels of self-esteem, body-image disturbance, and quality of life among this cohort. The data demonstrated that 46% presented borderline and moderate levels of severity of depression, while none had severe depression. It is worth noting that depression was seen in cases of severe to extensive hair loss. Body image concerns were reported by 11.6%, and 12% had low self-esteem. Sixty-two percent stated it had no impact on their quality of life [18]. Interesting data were presented by Muhaidat et al. that looked at the impact of AGA on the quality of life of 522 Jordanian individuals. A greater impact of AGA was observed among younger patients compared to older ones. Patients who were diagnosed with the disease for >10 years showed significantly milder effects [20]. Frith et al. conducted a systematic review and meta-analyses in order to explore the psychosocial impact of AGA on men. The study, which included data from 37 articles, showed a moderate impact on quality of life [21].

### 2.1. AGA Pathophysiology

Both genetic and androgenic factors play definite roles in the pathophysiology of the disease. Androgens can act by at least two pathways: usually, androgens bind to intracellular androgen receptors, which then translocate to the nucleus and bind to specific DNA sequences that regulate gene expression. Androgens can also bind to androgen receptors in the cytoplasm or at the plasma membrane, inducing the activation of a set of protein kinases, including proto-oncogene tyrosine-protein kinase (Src), ERK, protein kinase A (PKA), protein kinase B (Akt), and protein kinase C (PKC), as well as expression of downstream genes. Androgens can also increase intracellular levels of inositol-1,4,5-triphosphate, along with an increase in calcium influx and calcium-mediated cell signaling [14,22,23,24,25,26,27].

Androgens also alter the production of regulatory factors (soluble paracrine factors and extracellular matrix components) in the dermal papilla cells. It was found that the metabolite 5 α-dihydrotestosterone (DHT) particularly affects genetically susceptible cells of the dermal papilla, causing progressive hair follicle miniaturization and hair cycle abnormalities leading to male AGA [28,29]. The androgen receptor is found exclusively in the human dermal papilla cells of hair follicles and nowhere else in the surrounding tissue [28,29,30]. The 5α-reductase enzymatic activity produces the DHT that binds to the androgen receptor and activates it. The androgen receptors’ activity is manifested by the induction of a transactivation process in the dermal papilla cells. The direct binding of DHT to the androgen receptor in the dermal papilla cells of the hair follicle leads to miniaturization of the hair follicle. Indeed, elevated activity levels of the 5α-reductase, as well as increased presence of androgen receptor-DHT complexes, were found in the majority of AGA patients but not in non-AGA patients [31,32]. This suggests that suppressing the androgen’s receptor activity, either by producing less DHT due to decreased activity levels of the 5α-reductase or by interfering with the connection between the androgen receptor and DHT, could assist in preventing hair loss in AGA.

The crosstalk between the dermal papilla cells and the hair follicle cells results from the secretion of numerous growth factors and/or extracellular matrix factors from the dermal papilla cells [33]. These growth factors have an autocrine effect on the dermal papilla itself and a paracrine effect on the hair follicle epithelial cells. They include, for example, Insulin-like growth factor 1 (IGF-1). IGF-1 promotes proliferation, survival, and migration of hair follicle cells. In addition, IGF binding proteins also promote hair growth and hair cell survival by regulating IGF-1’s effects and its interactions with extracellular matrix proteins in the hair follicle. Furthermore, dermal papilla cells from non-balding scalps demonstrated significantly higher levels of IGF-1 compared to dermal papilla cells from balding scalps [34,35,36]. The PKC family of enzymes is also involved in androgen receptor desensitization, in modulating membrane structural events, in regulating transcription, in mediating immune responses, and in regulating cell growth. A significant increase in PKC-α levels in whole mouse skin at the mature anagen phase has been demonstrated. Overexpression of PKC-α in anagen hair follicles has also been reported during the natural growth of mouse hair [37]. The ERK and the Akt (also called protein kinase B-PKB) signaling pathways are the two major controllers of cell proliferation. The ERK signaling pathway plays key roles in the proliferation of many cell types, including dermal papilla cells [22,38]. Akt plays a critical role in mediating survival signals. Moreover, it was reported that the Akt pathway is also involved in regulating the survival of dermal papilla cells [39]. Vascular endothelial growth factor (VEGF), which is released from the epithelium, is a signaling protein that increases the vascular network surrounding the hair follicle. VEGF in dermal papilla cells regulates perifollicular vascularization as it increases significantly during the anagen phase and declines during the catagen and telogen phases. Hou et al. demonstrated that mRNA and protein expressions of VEGF fluctuated with a peak at anagen and a decrease at catagen to telogen [40,41,42].

The main factors that were found to play roles in the pathophysiology of AGA are presented in Table 1.

Additional aspects that might contribute to the pathogenesis of AGA include the fact that the skin acts as a neuro-immuno-endocrine organ, maintaining a functional role in hair growth [43]. For example, skin nerves can indeed modulate hair follicle development, growth, and/or cycling via the release of neurotransmitters, neuropeptides, and/or neurotrophins [44]. Melatonin is a neurohormone secreted by the pineal gland that regulates the mammalian circadian rhythm. Melatonin is also found in hair follicles [45]. The effect of melatonin treatment on hair growth was tested in clinical studies, which demonstrated a reduction in the degree of alopecia severity, as well as an increase in hair count and improvement in hair density and texture [46,47]. Dermal papilla cells express different cholinergic biomarkers. Hair growth was found to be regulated by the cholinergic system via muscarinic acetylcholine receptors. Bethanechol (a muscarinic agonist) increased hair shaft elongation in mice, thus demonstrating the cholinergic role in hair growth [48]. Thyroid-stimulating hormone receptors are expressed in various skin cell types, including melanocytes, keratinocytes, fibroblasts, and hair follicles. It has been shown that T4 stimulates the proliferation of hair follicle keratinocytes and that T3 inhibits their apoptosis [49]. Deficiency of vitamin D was found to be associated with alopecia areata, telogen effluvium, and AGA. Treatment with vitamin D has proven to be useful in the regrowth of hair within in vivo models. A literature review conducted by Zubair et al., which aimed to investigate the correlation between AGA and serum vitamin D levels, concluded that serum vitamin D might be a possible parameter for diagnosing the onset and severity of AGA [50].

### 2.2. Current Treatments of AGA

The goal of AGA treatments is to prevent the miniaturization process and, if possible, to reverse it. The therapeutic targets of AGA are to reduce DHT production, induce vasodilatory effects, trigger anagen, prolong anagen, and suppress inflammation. Thus, a variety of pharmacotherapeutic agents and procedural modalities that aim to achieve these goals are being tested.

Among the pharmacological therapies being studied are the following: Finasteride (inhibits the 5-alpha reductase type 2 enzyme) [51]; Minoxidil (probably induces vasodilatory- and angiogenesis-facilitating effects) [52]; prostaglandins (regulate the hair growth cell cycle) [53,54]; Valproic acid (promotes hair regeneration); and *Serenoa repens* (a type of palm that was found to induce a competitive non-selective inhibition the of androgen receptors types I and II with reduced DHT uptake by the hair follicle) [55,56]. Gupta et al. conducted a systematic search in PubMed, EMBASE, Scopus, and clinicaltrials.gov for randomized controlled trials investigating the efficacy of non-surgical AGA monotherapies. Platelet-rich plasma, low-level laser therapy, 0.5 mg Dutasteride, 1 mg Finasteride as well as 5% and 2% Minoxidil were found to be effective treatments for AGA in males. Low-level laser therapy and 5% and 2% Minoxidil demonstrated good responses in females [57]. It seems that the development of more efficient treatment modalities requires further validation by future randomized controlled trials.

It is worth mentioning that due to the increasing need for new, safe, and effective treatments for AGA, many off-label, topical monotherapies for treating AGA in men are suggested. The 2012 National Health Interview Survey reports concluded that complementary and alternative medicine approaches for dermatological conditions were used by 17.7% of Americans. The reports showed enhanced hair growth by using amino acids, caffeine, capsaicin, curcumin, garlic gel, marine proteins, melatonin, onion juice, procyanidin, pumpkin seed oil, rosemary oil, saw palmetto, vitamin B7 (biotin), vitamin D, vitamin E, and zinc [58]. A literature search was conducted to obtain randomized, controlled, and blinded studies that investigated off-label, topical monotherapies in male patients. Fourteen off-label topical therapies were investigated. Sixteen studies met the inclusion criteria, and a total of 1709 participants, aged on average 40 years, were included. Nine off-label therapies, which were mostly prostaglandin analogs and polyphenols such as Latanoprost and procyanidin oligomer, were reported to produce a significantly greater improvement in hair restoration parameters (e.g., mean change in hair count and hair diameter) as compared to placebo (*p* < 0.0001 at week 24) [59]. In recent years, a number of commercial devices using low-level laser therapy have been introduced. For example, in a randomized, sham device-controlled, double-blind clinical trial, a total of 128 male and 141 female subjects received either a laser comb or a sham device and were treated on the whole scalp three times a week for 26 weeks. A statistically significant difference in the increase in terminal hair density between laser-comb- and sham-treated groups was observed, and a higher percentage of laser-comb-treated subjects reported overall improvements in hair loss condition and hair thickness and fullness compared with sham-treated subjects [60]. Hair transplantation is an effective therapy using healthy follicular units to reestablish the hair-bearing scalp in appropriately selected patients [61]. Most of the available off-label therapies are readily available, but their efficacy and safety profiles are not as well established as those approved by the FDA. There is an increasing concern about the safety and efficacy of these off-label hair products.

However, despite the demand, there are only two FDA-approved medications to treat AGA: topical Minoxidil (5% and 2%) for men and women, and oral Finasteride (1 mg tablets- Proscar and Propecia) for men; these medications were approved in compliance with strict regulatory standards and have been extensively researched. FDA alerts health care providers, compounders, and consumers to potential risks associated with compounded topical Finasteride products. In a press release, the FDA stated it is aware of some compounders and telemedicine platforms that market topical formulations of Finasteride either as a single active ingredient (Finasteride alone) or in combination with other active ingredients, such as Finasteride combined with Minoxidil, to treat hair loss [62].

Both of these interventions have demonstrated some effectiveness at controlling AGA with long-term daily use; however, these are costly, require lifelong treatment, and may have side effects. Thus, the high demand for novel and more effective treatments for AGA remains valid.

## 3. Exploiting Different Molecular Mechanisms of Hair Growth Induced by Pharmacological Therapies for the Treatment of AGA

Some pharmacological therapies for AGA have been extensively studied as topical treatments for AGA, including Finasteride (DTH-suppressing 5α-reductase inhibitor), Latanoprost (a prostaglandin F2α analog), and Minoxidil (a pyrimidine derivative). Each of these substances affects hair growth through diverse mechanisms of action.

### 3.1. Finasteride

Finasteride was found to induce an anti-androgenic effect by altering the conversion of testosterone to DHT. It has already been shown that serum levels of DHT decrease upon topical administration of Finasteride to the bald area of AGA patients [63]. Thus, the topical application of Finasteride results in reduced peripheral DHT concentrations, as well as reduced levels of DHT in the dermal papilla cells, leading to decreased androgenic activation, and thereby reversing the follicular miniaturization of the hair. Biopsy specimens were collected from the balding and the non-balding areas of men with AGA, before and after 4 months of oral 1 mg Finasteride therapy. IGF-1 was upregulated by Finasteride treatment in 4 of 9 patients. Among the patients with increased IGF-1, three of them showed moderate clinical improvement after 12 months of treatment, and the other patient remained unchanged. In contrast, three patients with decreased IGF-1 expression in the balding scalp showed clinical worsening after 12 months. The other two patients without noticeable change in IGF-1 expression showed either slight improvement or no change in the condition of their hair [64]. The effect of Finasteride on stem cell properties was tested in a dermal papilla cell line and two human primary DP cells (HDPCs1 and HDPCs2). Finasteride significantly increased the levels of p-AKT at the concentrations of 10–100 μM. Moreover, in response to Finasteride treatment, activated AKT was increased together with an increase in cellular β-catenin, suggesting that Finasteride could maintain stem cell signaling through an AKT/β-catenin-dependent mechanism [65].

Dutasteride (0.5 mg), a selective inhibitor of both type 1 and type 2 5-alpha reductase enzymes, was found to be more effective in improving hair density and width of the shaft compared to Finasteride [66].

### 3.2. Minoxidil

Minoxidil was found to affect the androgen receptor in the dermal papilla cells by its ability to interfere with the binding of peptides and co-regulators to the androgen receptor, as well as disrupting the N-C interactions. It also acts as a chelating agent in different grooves on the androgen receptor, thereby reducing the stability of the receptor proteins. Hence, Minoxidil shows effectiveness in hampering the action of androgen receptor regulatory proteins, suppressing the downstream transactivation process [67,68]. Saline and 3% Minoxidil were topically applied twice daily for a total of 14 days on the dorsal skin of C57BL/6 mice after depilation. Increased expression of IGF-1 in Minoxidil-treated mice was observed compared to the saline-treated mice [69]. In an additional in vivo study, 3% Minoxidil was applied to the shaved dorsal area of C57BL/6, 6 days a week, for 4 weeks. Here again, IGF-1 expression levels were significantly higher in the dorsal skin derived from the Minoxidil-treated mice compared to those from the saline-treated mice [70]. Han et al. evaluated cell proliferation in cultured dermal papilla cells and measured the expressions of ERK, Akt, Bcl-2, and Bax (apoptosis regulator, also known as Bcl-2-like protein) upon 1.0 µM Minoxidil treatment. It was found that Minoxidil significantly increased the proliferation of the dermal papilla cells. The levels of ERK phosphorylation and of phosphorylated Akt increased significantly 1 h post-treatment. Moreover, a significant elongation of individual hair follicles in organ culture was observed after adding Minoxidil [59]. In another study, mouse hair follicles that were isolated on day 10 after depilation, and the bulge or dermal papilla regions were dissected. The bulge and dermal papilla cells were cultured with 100 μM Minoxidil for 10 days. Minoxidil influenced the bulge and the dermal papilla cells’ survival. In addition, Kras, Akt, ERK, Shh (gene regulating embryonic embryogenesis), and β-catenin mRNA levels were changed in response to Minoxidil treatment in both bulge and dermal papilla cells [71]. The effect of Minoxidil on VEGF protein synthesis in cell extracts and in dermal papilla cell-conditioned medium was investigated by Lachgar et al. Both the VEGF mRNA and protein were significantly elevated in treated dermal papilloma compared with controls. Dermal papilla cells were incubated with increasing Minoxidil concentrations (0.2, 2, 6, 12, and 24 mumol/L), showing a dose-dependent expression of VEGF mRNA. Similarly, VEGF protein production increased in cell extracts and conditioned media following Minoxidil stimulation [72]. These studies strongly support the involvement of Minoxidil in the development of dermal papilla vascularization via the stimulation of VEGF expression and support the hypothesis that Minoxidil has a physiological role in maintaining good vascularization of hair follicles in AGA [73]. Several studies have reported that potassium channel openers affect cultured follicles. Isolated red deer anagen follicles cultured in serum-free medium without streptomycin were treated with Minoxidil (0.1–100 µM). The drug increased the growth of the cells, while the potassium channel inhibitor, Tolbutamide (1 mM), inhibited cell growth and abolished the effect of 10 µM Minoxidil [74]. A recent study demonstrated that adenosine triphosphate (ATP) synthase, independent of its role in ATP synthesis, promotes stem cell differentiation. Minoxidil was found to increase the amount of intracellular Ca^2+^, which has been shown to upregulate ATP synthase [75]. Minoxidil caused an increase in Ca2+ levels and VEGF production in cultured DPC [76].

### 3.3. Latanoprost

The addition of Latanoprost to human ciliary muscle cell cultures increased ERK1/2 activity. Interestingly, the latanoprost-induced ERK1/2 activation was blocked by the presence of PKC inhibitors and by downregulation of PKC through prolonged incubation with a phorbol ester [77]. Latanoprost increased angiogenesis by elevating VEGF levels compared to the PBS control group in the chick CAM model in vivo [78]. Zhang et al. incubated human dermal papilla cells with five different concentrations of Latanoprost for 72 h, and then VEGF protein synthesis was investigated. VEGF protein synthesis was significantly elevated in a dose-dependent manner in the Latanoprost-treated dermal papilla cells compared with the controls, except the 0.16 ng/mL Latanoprost-treated group [79].

Bimatoprost, prostaglandin F2 analogs (0.03% solution), approved for eyelash hypotrichosis (much like Latanoprost) and are known to induce the anagen phase in hair follicles, have been tested in clinical trials that reported that the drug was effective compared to placebo [53].

The effects of Finasteride, Latanoprost, and Minoxidil on the various mechanisms of action that are involved in the pathogenesis of AGA are presented in Figure 2.

### 3.4. Additional Drugs Aimed at Targeting the Various Mechanisms of AGA Pathogenesis

Other drugs targeting the various mechanisms of AGA pathogenesis are being tested. Cetirizine, H1 antihistaminic, and a prostaglandin D2 production reducer were tested in a clinical study of 60 subjects. The treatment resulted in significantly greater hair growth and patient satisfaction compared to controls [80]. SM04554, JW0061, and KY1938, Wnt pathway modulators were found to promote hair growth and hair follicle proliferation, and hair regeneration [81,82,83]. Valproic acid and Cyclosporine A, known to activate the canonical Wnt/β-catenin pathway, have also been shown as enhancers of the hair growth effect [84]. Subcutaneous administration of autologous platelet-rich plasma is currently utilized in the treatment of AGA. Alpha granules of platelets contain VEGF [85]. Additionally, its effects include hypoxia reduction, vasodilation, vasoconstriction, and inflammation in bald areas while promoting neo-angiogenesis [86].

Thus, it seems that each of the above drugs affects hair growth by a direct (anti-androgenic effect and prompting cell growth) as well as by an indirect (increasing oxygen, nutrition, and blood supply to the dermal papilla cells and increasing absorption and penetration of drugs) mechanisms of action, playing a role in the prevention of AGA. Not only that, it could be suggested that a combination of the drugs could activate the various mechanisms simultaneously to induce a synergistic effect on hair growth.

## 4. Combined Topical Treatments for AGA

Indeed, a combined topical treatment of the drugs has been tested in some human clinical studies.

### 4.1. Combination of Finasteride + Minoxidil in the Treatment of AGA

In a prospective pilot study, Rafi et al. assessed the efficacy of four regimens for the treatment of AGA in 15 atopic and nonatopic patients. The first one was NuH Hair, which is a research product formulation, administered topically, consisting of the combination of Finasteride, Dutasteride, and Minoxidil blended into a hypoallergenic lotion. The other three treatments included a commercial 5% *Minoxidil Foam* (Rogaine), 1 mg *Finasteride tablet* (Propecia), and ketoconazole shampoo. Each patient served as their own control. Significant hair growth was reported by all patients. In those patients who utilized all four components, significant hair growth was demonstrated within 30 days. In those patients who were treated only with NuH Hair, significant growth was demonstrated after 3 months. The researchers concluded that a combined treatment achieves significant and rapid growth of new hair [87].

Tanglertsampan conducted a study aimed at comparing the efficacy and safety of the 6-month application of 3% Minoxidil lotion versus combined 3% Minoxidil and 0.1% Finasteride lotion in 40 men with AGA. At the end of month six, hair counts had increased from baseline in both groups. However, a paired *t*-test revealed a statistical difference only in the 3% Minoxidil and 0.1% Finasteride lotion group (*p* = 0.044). An unpaired *t*-test revealed no statistical difference between the two groups with respect to the change in hair counts at 6 months from baseline (*p* = 0.503). Using global photographic assessment, the 3% Minoxidil and 0.1% Finasteride lotion group demonstrated a significantly higher efficacy than the 3% Minoxidil group (*p* = 0.003). There was no significant difference in side effects between the two groups. Thus, the author suggested that since the significant improvement in the 3% Minoxidil and 0.1% Finasteride lotion group was greater than the 3% Minoxidil group, and no serious adverse events occurred (including no sexual side effects in either group), the combination may be safe and effective as a treatment option for AGA [88].

In another study, the authors evaluated the maintenance of hair growth by treatment with 5% topical Minoxidil in combination with 0.1% topical Finasteride in 50 patients with AGA. All patients had been initially treated with topical Minoxidil and oral Finasteride for a period of two years, after which the oral Finasteride was replaced with topical Finasteride. Five of fifty patients had discontinued the treatment for a period of 8–12 months and then resumed with combined topical Minoxidil and Finasteride. Of the 45 patients (on continuous treatment), 25 patients’ hair density was moderately maintained. When shifting from oral to topical Finasteride, these patients initially experienced some hair loss and then reached a plateau phase. They had no further hair loss, and hair density was well maintained. Seven patients had a slight decline in hair density when switching from oral Finasteride to topical treatment, and thirteen patients did not experience a decline and even showed an improvement in hair density. Five of the fifty patients had stopped all treatment for hair loss and showed a decrease in hair density over a period of 8–12 months after discontinuing treatment. All these patients improved when they started treatment with topical Minoxidil and topical Finasteride. Four of these patients demonstrated an increase in hair density at the one-year follow-up. Overall, data from the study revealed that 84.44% of patients maintained hair density well, displaying the effectiveness of the combination in maintaining hair growth. In five patients who had discontinued the treatment, it was noted that 80% of them showed a substantial improvement within a year upon restarting the treatment with topical Finasteride combined with Minoxidil. The combination of the drugs was well tolerated, with no psychological fear of oral medication and good compliance [89].

Sheikh et al. conducted preclinical toxicity studies as well as a clinical study to examine the safety and efficacy of a combined formulation of 5% Minoxidil + 0.1% Finasteride lipid solution in comparison with a 5% Minoxidil solution. For the preclinical toxicity studies, Sprague Dawley Albino rats were shaved ~24 h before the first dose’s application. A volume of 1 mL of placebo, 1% Minoxidil, and 0.2% Finasteride (low dose), 3% Minoxidil and 0.6% Finasteride (medium dose), and 5% Minoxidil and 1% Finasteride (high dose), used as the dosing solutions, were applied topically, drop by drop, over the shaven area of each rat twice daily for 28 consecutive days. All rats were free of signs of toxicity throughout the study period. There was also no noticeable difference in serum clinical chemistry values between the treatment groups. The histopathology of the liver, kidney, heart, and spleen indicated no significant pathological changes. Thus, all tested doses were safe and well tolerated. In the clinical study, the patients were randomized to receive topically either a 5% Minoxidil + 0.1% Finasteride lipid solution or 5% Minoxidil alone for 6 months. It was observed that a significantly higher number of patients treated with the combination of 5% Minoxidil + 0.1% Finasteride exhibited a greater improvement in the investigator score compared to those treated with Minoxidil alone (64.7% in the 5% Minoxidil + 0.1% Finasteride lipid solution group vs. 25.5% in the Minoxidil-alone group; *p* = 0.0006). Global photographic assessment revealed a statistically significant difference between the groups. More patients who were treated with 5% Minoxidil + 0.1% Finasteride demonstrated a greater improvement (i.e., “lightly”, “moderately”, and “greatly” increased) in hair growth score as compared to those treated with Minoxidil alone (88.9% in the 5% Minoxidil + 0.1% Finasteride solution group vs. 60.0% in the Minoxidil-alone group). Patients’ self-assessment score was also determined. Significantly more patients treated with 5% Minoxidil + 0.1% Finasteride displayed a “strongly agree” and “agree” response to the bald area becoming smaller for the hair growth and loss of hair growth questionnaire as compared to the 5% Minoxidil group (*p* = 0.0008). The majority of patients did not have any adverse event during the trial with 5% Minoxidil + 0.1% Finasteride, clinically significant abnormalities during laboratory assessments, vital recordings, ECG recordings, etc. No deaths or serious adverse events were reported during the course of the trial [90].

To compare the efficacy and safety of a topical solution of 0.25% Finasteride mixed with 3% Minoxidil vs. 3% Minoxidil solution, 40 men with AGA were randomized into two groups: 6 months of treatment twice daily with a Finasteride/Minoxidil combination, or a Minoxidil solution. At the end of the study, the combined solution of Finasteride and Minoxidil displayed significantly superior results compared to that of Minoxidil alone regarding improvements in hair density, hair diameter, and global photographic assessment by patients and by the investigators (all *p* < 0.05). About 90% of patients treated with the combined solution experienced moderate to marked improvement. The combined solution also had a minimal effect (an approximately 5% reduction) on plasma DHT levels. There were no systemic adverse events reported by patients in both groups [91].

A year later, the same group published data from a 6-month, randomized, double-blind, controlled study aimed at evaluating the efficacy of the combination of 0.25% topical Finasteride and 3% Minoxidil versus 3% Minoxidil solution in females with AGA. By the end of the treatment schedule, hair density and diameter had increased in both groups. However, the Finasteride + Minoxidil group was significantly superior to the Minoxidil-alone group in terms of hair diameter (*p* = 0.039). No systemic side effects were reported. Serum DHT levels in the Finasteride/Minoxidil group were significantly decreased from baseline (*p* = 0.016) [85].

Marotta et al. investigated the clinical efficacy and patient satisfaction of a topical compounded formulation (Minoxidil 0.1% Finasteride, 0.2% biotin, and 0.05% caffeine citrate hydroalcoholic solution) in five male AGA patients. Patients were provided with the topical formulation and instructed to apply a 1 mL dose to the entire frontal, parietal, and occipital scalp, twice daily for 6 months. By the end of the study, the investigators concluded that the topical treatment was successful for all patients. Although moderate, the clinical improvements were visually noticeable as most patients had thicker, more voluminous hair, improved scalp coverage, and improved general hair appearance. These results were consistent with the photographic assessment, which demonstrated a global average increase of +1.05 in the patients’ hair density. According to the patient self-assessment, the topical compounded formulation was effective following 3 and 6 months of continuous treatment [86].

Interestingly, Oliveira et al. noticed that the combination of Minoxidil and Latanoprost is currently emerging as a promising strategy for the treatment of AGA. Thus, they developed an efficient LC–MS bioanalytical method to simultaneously quantify Minoxidil and Latanoprost levels in different skin layers. An in vitro skin penetration study, using an aqueous solution containing Minoxidil sulphate and Latanoprost as formulation, was performed. At the end of 24 h of treatment, the concentration of the drugs that penetrated each of the skin’s layers was calculated. Minoxidil sulphate permeated the stratum corneum and accumulated in deeper skin layers after 24 h of exposure. Latanoprost, on the other hand—with a much higher partition coefficient than that of Minoxidil sulphate—was retained in the stratum corneum, which is the skin layer with the most lipophilic characteristics, and had difficulty diffusing to the innermost skin layers that are more hydrophilic [87].

A summary of the data from clinical studies demonstrating the efficacy and safety of the combined topical treatment of Finasteride and Minoxidil are presented in Table 2.

### 4.2. Combination of Latanoprost + Minoxidil in the Treatment of AGA

Bloch et al. conducted a study in which they evaluated the efficacy of Minoxidil versus Latanoprost or combined to treat 98 patients with AGA. This was a double-blind, comparative study that included six treatment groups: placebo; topical lotion containing 5% Minoxidil; topical lotion containing 5% Minoxidil + 0.005% Latanoprost; topical lotion containing 0.005% Latanoprost; topical lotion containing 5% Minoxidil + 0.010% Latanoprost and topical lotion containing 0.010% Latanoprost. The products were applied daily by topical application on the dry and clean scalp, using fingertips. The duration of the treatment was 6 months for all participants, except for the group treated with 5% Minoxidil + 0.005% Latanoprost, whose treatment was prolonged for an additional 2 months. Comparative visual analysis of the macro images revealed that there was no improvement in the placebo and in the 0.010% Latanoprost groups compared with the baseline. However, a visual improvement was noted in the various groups—35% of participants in the 5% Minoxidil, 36% of the participants in 5% Minoxidil + 0.005% Latanoprost, 19% of the participants in 0.005% Latanoprost, and 6% of the participants in the 5% Minoxidil + 0.010% Latanoprost groups in comparison to the baseline. A statistically significant increase in the total number of hairs and anagen hair count from baseline was observed in all the treatment groups except for the placebo and the 0.010% Latanoprost groups. None of the six treatments significantly reduced the amount of telogen hairs. Overall, the 0.005% Latanoprost treatment yielded slightly inferior results to 5% Minoxidil in the total increase in number of hairs and in the total number of anagen hairs after 180 days of use, while 0.010% Latanoprost did not yield a statistically significant improvement, with results comparable to the placebo group. The treatment with topical 5% Minoxidil increased the total number of hairs and the total number of anagen hairs in the first 3 months of the study, while treatment with topical 0.005% Latanoprost lotion or treatment with topical 5% Minoxidil 5% + 0.010% Latanoprost lotion increased the total number of hairs after 6 months of use. No discomfort, pruritus, scaling, or erythema related to the treatments and no clinical signs were detected on the scalp during the whole study duration as reported by the participants. Thus, the products were well tolerated [88]. A summary of clinical studies demonstrating the efficacy and safety of combined topical treatments of Latanoprost + Minoxidil is represented in Table 3.

## 5. Additional Combined Topical Treatments for AGA

### 5.1. Combined Topical Treatment for AGA with a Combination of Minoxidil, Finasteride, and Latanoprost

Topical treatment for AGA with a combination of the three drugs has also been studied. A mixture of Minoxidil 5%, Finasteride 0.1%, and Latanoprost 0.03%, identified as the topical product TH07, was used with once-daily application in a randomized, double-blind comparative study that included 40 men between the ages of 24 and 65 with AGA. Moderate hair re-growth was observed in the majority of participants treated with TH07 in comparison to the retreatment with its active components administered as monotherapy. The investigators’ assessment of hair growth was conducted using pictures taken from each patient before and at the end of the treatment. In the TH07 treatment group, 52% of the patients demonstrated a dense level (defined as sufficient growth to be cut and combed) of hair growth and 30% moderate growth (meaning that there was a visible change). No hair growth was observed in most of the subjects in the Finasteride, Latanoprost, and Minoxidil treatment groups (50%, 77%, and 75%, respectively). No systemic adverse events were reported, and the TH07 product was well tolerated [89]. A representative picture of one of the patients, before and after the treatment with TH07, is presented in Figure 3.

### 5.2. Additional Topical Drug Combinations for the Treatment of AGA

Additional drug combinations have also been reported. For example, Ketoconazole, a topical antifungal shampoo used for the treatment of seborrheic dermatitis, was found to enhance Finasteride’s effect by decreasing DHT levels in the male scalp [90]. A combined treatment of topical Cetirizine [a prostaglandin D2 receptor antagonist) and topical Minoxidil was tested in females with AGA. An increase in the hair shaft thickness and greater clinical improvement was demonstrated by the combination compared to topical Minoxidil with placebo [91]. Interestingly, the mechanism of action involved in microneedling, a widely used technique in dermatology, is based on the regenerative activation and stimulation of bulge stem cells (a main type of hair-follicle stem cell involved in hair growth), inducing overexpression of VEGF, β-catenin, and Wnt pathway products [92]. It was observed in various studies that Minoxidil with microneedling was more effective than Minoxidil alone. Abdi et al. conducted a systematic review and meta-analysis of ten randomized controlled trials, including 466 patients, to determine the efficacy of combination therapy using topical Minoxidil with microneedling compared to topical Minoxidil treatment alone. The data collected and analyzed indicated that the use of microneedling in combination with topical Minoxidil for the treatment of AGA has yielded positive results, both qualitatively and quantitatively [97].

### 5.3. Satisfaction of Patients with AGA from Topical Treatment with TH07

The satisfaction of patients with AGA from topical treatment with TH07 was evaluated in an additional study. The data revealed that most of the patients treated with TH07 were satisfied with their hair appearance compared to the other treatments. They were highly satisfied with the efficacy and tolerability of the treatment, and expressed interest in continuing with it. They reported that the drug was easy to handle and showed a high rate of compliance with the treatment. Interestingly, patients felt that the current treatment was more effective and resulted in fewer side effects than the topical and systemic previous treatments they had for AGA [98].

## 6. Conclusions

The information about the mechanisms of action of the various drugs used in the treatment of AGA, together with exploration of the mechanisms involved in the pathogenesis of the disease, has led to the insight that those mechanisms of action can be used to treat the disease. Furthermore, the data from the combined treatment of the drugs indicate that activating or suppressing several mechanisms of action by treating simultaneously with a combination of the drugs resulted in a good and safe therapeutic outcome. Thus, it is recommended that the correlation between the pathogenesis of AGA and the mechanisms of action of drugs that are developed for the treatment of the disease should be taken into consideration in order to achieve more effective and safer treatments.

## Figures and Tables

**Figure 1 ijms-26-10712-f001:**
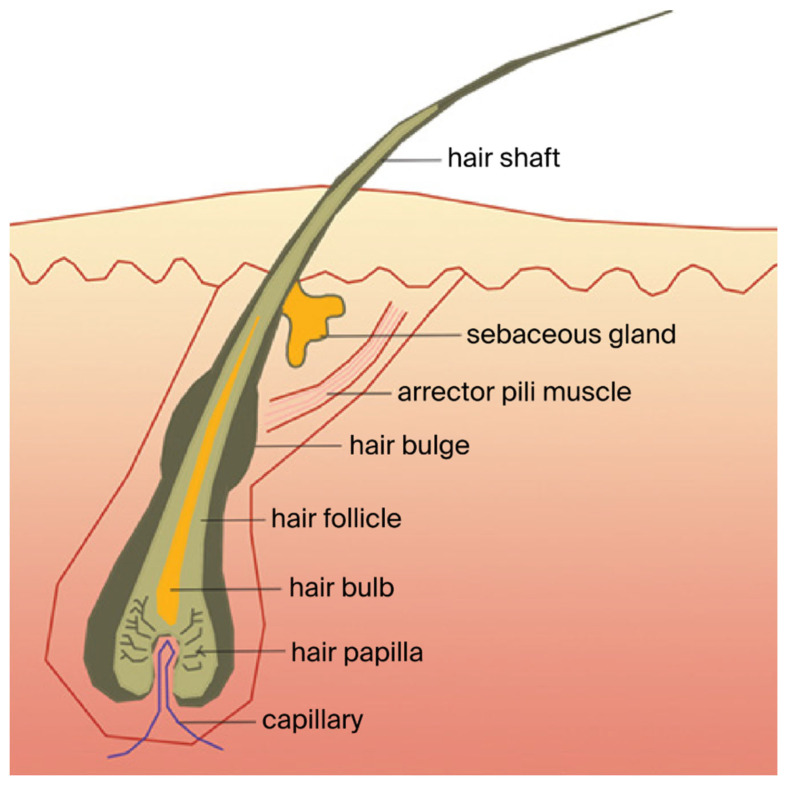
Diagram of an anagen follicle. (Erdoğan, 2017. *Hair and Scalp Disorders*, IntechOpen. DOI: 10.5772/67269 [7]).

**Figure 2 ijms-26-10712-f002:**
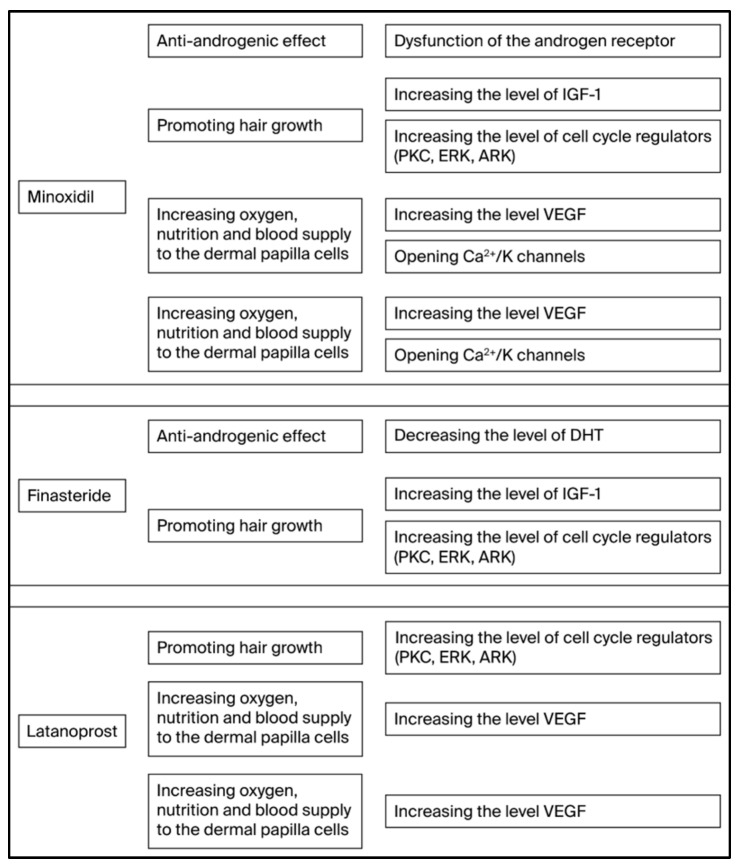
The effect of the Finasteride, Latanoprost, and Minoxidil on the various mechanisms of action that are involved in the pathogenesis of AGA.

**Figure 3 ijms-26-10712-f003:**
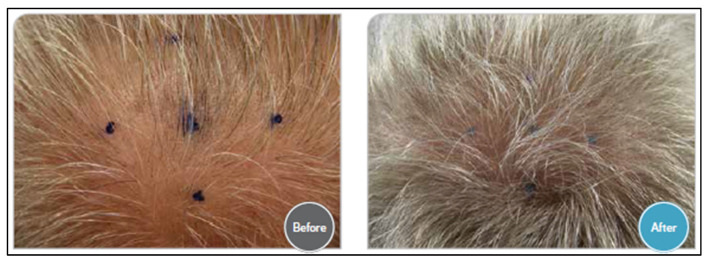
A representative picture of one of the patients, before and after treatment with TH07.

**Table 1 ijms-26-10712-t001:** The main factors that were found to play roles in the pathophysiology of AGA.

Androgen Receptors	Activated by 5-Dihydrotestosterone	Induces Progressive Hair Follicle Miniaturization and Hair Follicle Abnormalities
Growth factors and/or extracellular matrix factors	Insulin-like growth factor 1 (IGF-1)	Promotes proliferation, survival, and migration of hair follicle cell, leading to hair growth
	PKC Family	Promotes androgenic receptor desensitization, mediating immune responses and regulating dermal papilla cell growth
	ERK and Akt	Regulate the survival of dermal papilla cells
	VEGF	Regulates perifollicular vascularization of dermal papilla cells

**Table 2 ijms-26-10712-t002:** Clinical studies demonstrating the efficacy and safety of combined topical treatments of Finasteride and Minoxidil.

Origin	No. of Patients	Treatments	Assessments	Study Outcomes	Adverse Events
Rafi et al. (2011) [88]	15 Males	A lotion including Finasteride, Dutasteride, and Minoxidil with Minoxidil Foam 5% or Finasteride tablet 1 mg or ketoconazole shampoo for 9 months.	Hair growth.	All patients demonstrated significant growth of hair. In those patients who utilized all 4 components, significant growth was achieved in as little as 30 days. In those patients who choose only to utilize combination of Finasteride, Dutasteride, and Minoxidil, significant growth was demonstrated after 3 months.	No side effects were reported.
Tanglertsampan et al. (2012) [89]	40 Males	Minoxidil lotion 3% vs. combined Minoxidil 3% and Finasteride lotion 0.1% for 6 months.	Hair counts and global photographic assessment.	No statistical difference between two groups with respect to change in hair counts at 6 months from baseline. Global photographic assessment revealed that Minoxidil 3% and Finasteride lotion 0.1% group demonstrated a significantly higher efficacy than Minoxidil 3% group.	No significant difference in side effects between both groups.
Chandrashekar et al. (2015) [90]	50 Males	Topical Minoxidil 5% and topical Finasteride 0.1% for 12 months, after initial treatment with topical Minoxidil and oral Finasteride for two years.	Hair density.	Patients who were shifted from oral to topical Finasteride maintained their hair density. Five patients stopped all treatment for a year, and a decrease in hair density over a period of 8–12 months after stopping treatment was noted, but improvement was observed upon treatment with topical Minoxidil and topical Finasteride.	Combination of drugs was well tolerated.
Sheikh et al. (2015) [91]	50 Males	Topical Minoxidil 5% and Finasteride solution 0.1% vs. Minoxidil 5% for 6 months.	Investigators’ assessment, global photographic assessment, and patients’ self-assessment.	Patients who were shifted from oral to topical Finasteride maintained their hair density. Five patients stopped all treatment for a year, and a decrease in hair density over a period of 8–12 months after stopping treatment was noted, but improvement was observed upon treatment with topical Minoxidil and topical Finasteride.	Majority of patients did not have any adverse events during trial.
Suchonwanit et al. (2018) [92]	40 Males	Topical Finasteride 0.25% and Minoxidil solution 3% vs. topical Minoxidil solution 3% for 6 months.	Hair density, hair diameter, and global photographic assessment.	Combined solution of Finasteride and Minoxidil was significantly superior to Minoxidil alone in improvements in hair density, hair diameter, and global photographic assessment by patients and by investigators.	There were also no systemic adverse events reported by patients in both groups.
Suchonwanit et al. (2019) [93]	30 Females	Topical Finasteride 0.25% and Minoxidil solution 3% vs. topical Minoxidil solution 3% for 6 months.	Hair density and hair diameter.	Finasteride and Minoxidil group was significantly superior to Minoxidil group in terms of hair diameter.	No systemic side effects were reported.
Marotta et al. (2020) [94]	5 Males	Topical Minoxidil 10%, Finasteride 0.1%, biotin 0.2%, and caffeine citrate 0.05%, hydroalcoholic solutions, for 6 months.	Hair thickness, vellus hair counts, scalp coverage, general hair appearance, photographic assessment, patients’ self-assessment.	Most patients had thicker, more voluminous hair, improved scalp coverage, and improved general hair appearance. Increase of +1.05 in patients’ hair density was observed. According to patients’ self-assessment, topical compounded formulation was effective.	No side effects were reported.

**Table 3 ijms-26-10712-t003:** Data from clinical studies demonstrating the efficacy and safety of combined topical treatments of Latanoprost + Minoxidil.

Origin	No of Patients	Treatments	Assessments	Study Outcomes
Bloch et al. (2018) [95]	98 participants	Placebo; 5% Minoxidiltopical lotion; 5% Minoxidil + 0.005% Latanoprost; 0.005% Latanoprost; 5% Minoxidil + 0.010% Latanoprost; and 0.010% Latanoprost.	Comparative visual analysis of the macro images.	All lotion products were well tolerated Statistically significant increases in the total number of hairs and anagen hair count from baseline were observed in all treatments except placebo and 0.010% Latanoprost None of the treatments significantly reduced the amount of telogen hairs
Sekhavat et al. (2023) [96]	40 men	Topical solution, a mixture of Minoxidil 5%, Finasteride 0.1%, and Latanoprost 0.03%, defined as TH07.	Investigators’ assessment, global photographic assessment, and patients’ self-assessment.	Most of the patients treated with TH07 were satisfied with their hair appearance. 52% of the patients demonstrated a dense level 30% moderate growth; 18% mild growth

## Data Availability

No new data were created or analyzed in this study. Data sharing is not applicable to this article.

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
