# Peer review of "Using the Mechanisms of Action Involved in the Pathogenesis of Androgenetic Alopecia to Treat Hair Loss"

_ijms, 2025, doi:10.3390/ijms262110712_

Round 1

Reviewer 1 Report

Comments and Suggestions for Authors

This manuscript provides a detailed and comprehensive overview of pharmacological interventions and associated molecular factors in AGA, offering valuable insights for specialist readers. However, there is noticeable redundancy and a lack of structural clarity in some sections. Furthermore, given the increasing number of non-specialist and international readers, the readability must be improved through clearer organization and the incorporation of visual aids such as figures or summary tables. Below are specific comments for revision:

Reviewer Comment 1: Eliminate redundancy between Introduction and Section 2.1
There is considerable overlap between the latter part of the Introduction and the beginning to middle of Section 2.1, particularly concerning growth factors such as IGF-1 and VEGF.
Clarify the respective roles of each section and eliminate redundant descriptions.
In the Introduction, restrict the content to physiological roles of growth factors under normal conditions, while Section 2.1 should focus on how these factors are altered or dysregulated in AGA. Additionally, create a figure or table summarizing key factors in each hair cycle phase to enhance understanding.

Reviewer Comment 2: Clearly state the epidemiological and social significance of AGA
As AGA is an age-related condition, its epidemiological position within the broader spectrum of alopecias should be concisely introduced in the early part of the manuscript.
Briefly explain the increasing relevance of AGA in the context of aging societies and its impact on quality of life (QOL). This will help clarify the social significance and the rationale behind this review.

Reviewer Comment 3: Clarify the regulatory status of treatments and balance discussion of off-label therapies
In Section 2.2, explicitly state at the beginning that Minoxidil and Finasteride are the only FDA-approved treatments for AGA.
While the inclusion of various off-label therapies is informative, note that their long-term safety and efficacy are not yet well-established to provide a more balanced perspective.

Reviewer Comment 4: Accompany drug descriptions with levels of evidence
The classification by mechanism of action in Section 3 is appreciated. However, in areas where multiple studies are cited, include brief notes on the strength of evidence, such as recommendation levels or the presence/absence of meta-analyses.
This will enhance the clinical value of the information provided.

Reviewer Comment 5: Integrate Dutasteride into the Finasteride subsection
As Dutasteride shares the same pharmacological class (5α-reductase inhibitors) and clinical indications as Finasteride, merge the description of Dutasteride into the Finasteride subsection for better logical flow and reader convenience.

Reviewer Comment 6: Combine Latanoprost and Bimatoprost into a unified section
Since Latanoprost and Bimatoprost share similar mechanisms of action and evidence types, begin the section with Bimatoprost and then compare the two agents within a single, cohesive paragraph.
This restructuring will reduce redundancy and improve clarity.

Reviewer Comment 7: Organize studies by similarity and present key comparisons in a table
In Section 4, group studies with similar content or interventions together, and describe key differences among them. In particular, summarize the studies in Section 4.2 in a comparative table to facilitate cross-study evaluation and improve overall readability.

Reviewer Comment 8: Consolidate combination therapy discussion and prioritize objectivity
The content on triple combination therapy (Minoxidil + Finasteride + Latanoprost) in Section 5 appears somewhat disconnected and is largely based on subjective outcomes.
Integrate Section 5 into Section 6, and restructure the combined section so that objective data (e.g., hair count, thickness) are prioritized, while subjective measures (e.g., patient satisfaction) are briefly summarized.

Comments on the Quality of English Language

English is good.

Author Response

Please refer to the attached document, thank you.

Reviewer 2 Report

Comments and Suggestions for Authors

This is a well-structured and informative review article on the pathophysiology of androgenetic alopecia (AGA) and the mechanisms of action of various pharmacological treatments. The authors have effectively compiled a significant amount of data from both in vitro and in vivo studies, as well as clinical trials, to support the potential of both monotherapy and combination therapy. The article's main strength is its detailed focus on the molecular pathways affected by different drugs. However, several areas need to be addressed to enhance the document's professional quality and scientific rigor.

Figure 1 and Figure 2: The document mentions "Figure 1. Diagram of an anagen follicle" and then later says "The effect of the Finasteride, Latanoprost and Minoxidil on the various mechanisms of action... are presented in Figure 1". This is a major error in figure numbering. The first figure is a diagram of a hair follicle, and the second is a flowchart of drug effects. These should be labeled distinctly (e.g., Figure 1 and Figure 2).

Content of Figure 2: The flowchart labeled "Figure 2" on pages 14-15 is difficult to read due to poor image quality and overlapping boxes. More critically, the Latanoprost section is duplicated, and the "Promoting hair growth" section for Finasteride is also duplicated. A clear, high-resolution, and correctly formatted figure is essential to convey this complex information effectively.

The phrase "using the mechanisms of action" in the title is slightly awkward. A more professional phrasing would be "Exploiting the Mechanisms of Action..." or "Harnessing the Mechanisms of Action..."

There is a mix of formal and informal language. For example, "Hair loos"  is used instead of "hair loss." Another example is the phrase "which" at the end of a sentence. These need to be corrected throughout the document.

The abbreviation "DTH" is used in the keywords and later defined as DHT (dihydrotestosterone), but the abbreviation is not explicitly defined in the abstract. This should be addressed for clarity.

The description of the hair follicle structures is slightly confusing. A clearer, more sequential description would be beneficial.

Page 2: "expressed by to progressive miniaturization"  should be "expressed by progressive miniaturization."

Page 2: "which" at the end of the sentence should be removed.

Page 6 and 12: "protein kinae B- PKB" should be "protein kinase B- PKB".

Page 15: "effects hair growth by a direct (anti- adrogenic effect and prompting cell growth) as well as by an indirect"  should be "affects hair growth by a direct (anti-androgenic effect and promoting cell growth) as well as by an indirect."

Author Response

We thank the reviewer for his constructive comments on our submitted manuscript, entitled:” Using the Mechanisms of Action Involved in the pathogenesis of Androgenetic Alopecia to Treat Hair Loss”. Manuscript ID: ijms-3747417.

 We have tried our best to respond and upgrade the article accordingly.

The added text is highlighted in yellow.